

# Thermal stratification and meromixis in four dilute temperate zone lakes

Elizabeth D. Swanner[1][*], Chris Harding[1], Sajjad A. Akam[1], Ioan Lascu[2], Gabrielle Ledesma[1, 3], Pratik Poudel[1, 4], Heeyeon Sun[1, 4], Samuel Duncanson[1, 4], Karly Bandy[4], Alex Branham[4], Liza Bryant-Tapper[4], Tanner Conwell[4], Omri Jamison[4], and Lauren Netz[4].

1Department of Geological & Atmospheric Sciences, Iowa State University, Ames, IA, 50010, USA

2Department of Mineral Sciences, National Museum of Natural History, Smithsonian Institution, Washington, DC, 20560, USA

3Department of Earth Sciences, Memorial University of Newfoundland, St. John's, Canada

4Students in GEOL 406/506, Spring 2022, Iowa State University, Ames, IA, 50010, USA

Correspondence to: Elizabeth D. Swanner (eswanner@iastate.edu)

**Abstract.** Four adjacent lakes (Arco, Budd, Deming, and Josephine) within Itasca State Park in Minnesota, USA are reported to be meromictic in the scientific literature. However, seasonally persistent chemoclines have never been documented. We collected seasonal profiles of temperature and specific conductance and placed temperature sensor chains in two lakes for ~ 1 year to explore whether these lakes remain stratified through seasonal mixing events, and what factors contribute to their stability. The results indicate that all lakes are predominantly thermally stratified and are prone to mixing in isothermal periods during spring and fall. Despite brief, semi-annual erosion of thermal stratification, Deming Lake showed no signs of complete mixing from 2006 to 2009 and 2019-2022 and is likely meromictic. Geochemical data indicate that water in Budd Lake, the most dilute lake, is predominantly sourced from precipitation. The water in the other three lakes is calcium-magnesium bicarbonate type, reflecting a source of water that has interacted with the landscape. $\delta^{18}O_{H2O}$ and $\delta^{2}H_{H2O}$ measurements indicate the lakes are supplied by precipitation modified by evaporation. The water residence time in meromictic Deming Lake is short (100 days), yet it maintains a large reservoir of dissolved iron. Josephine, Arco, and Deming lakes sit in a valley with likely permeable sediments and may be hydrologically connected through wetlands, and recharged with shallow groundwater, as no streams are present. All four lakes develop subsurface chlorophyll maxima layers during the summer. All lakes also develop subsurface oxygen maxima that may result from oxygen trapping in the spring by rapidly developed thermoclines. Documenting the mixing status and general chemistry of these lakes enhances their utility and accessibility for future biogeochemical studies.

## 1 Introduction

Anoxia and hypoxia are common features of lakes (Nürnberg, 1995), and meromixis is an end-member of lake mixing regimes in which a lake does not mix seasonally, resulting in a permanently anoxic monimolimnion. Seasonally persistent





anoxic conditions, and even meromixis, are becoming more common as a result of climate change (Ficker et al., 2017), land-use changes (Jenny et al., 2016), and road salt contamination (Koretsky et al., 2012). Meromictic lakes are characterized by a sharp increase in electrical conductivity at a chemocline separating a cold and/or dense monimolimnion from the epilimnion

(Boehrer and Schultze, 2008). The epilimnion is in contact with the atmosphere, while the monimolimnion is anoxic. These conditions restrict the habitat of aerobic organisms, including zooplankton and fish, and can amplify microbial or geochemical cycling of elements across the redoxcline (Busigny et al., 2016) and impact the emission of greenhouse gases such as methane (Lambrecht et al., 2020).

The stability of a lake against mixing is conferred by density differences between the epilimnion and monimolomnion,

driven by the thermocline and chemocline. Lake morphometry is a key factor in the development of meromixis and can be useful to evaluate the potential for a lake to develop meromixis. Lakes that are relatively deep compared to their surface area, which determines the fetch upon which wind can act, are less likely to overturn. The potential for water column stratification is reflected by relative depth ($Z_r$; %), which is the ratio of the maximum lake depth ($Z_m$; m) to the diameter of a circle of area equal to that of the lake ($A_0$; $m^2$), expressed as a percentage in eq. 1:

$$Z_r = \frac{Z_m}{2}\sqrt{\frac{\pi}{A_0}} \times 100 \tag{1}$$

Most temperate zone lakes prone to meromixis have relative depths exceeding 4% and $A^0$ of less than 500,000 $m^2$ (Swanner et al., 2020). Density differences leading to the development of meromixis can also result from increased chemical constituents that enrich the water density of monimolimnion, such as from runoff, groundwater inputs, and remineralization processes associated with organic carbon loading (Hakala, 2004). Thermal stratification is typically not the main mechanism stabilizing

meromixis in lakes of the temperate zone (Boehrer et al., 2017).

Gradients of dissolved oxygen and inorganic nutrients are mediated by primary productivity and uptake in the epilimnion and sinking and decomposition of biomass in the monimolimnion (Camacho, 2006). Stratification limits the return of nutrients to the epilimnion. A subsurface chlorophyll maximum layer (SCML) often develops within the metalimnion, the middle seasonally mixed layer of meromictic lakes (Baker and Brook, 1971; Camacho, 2006). Stable stratification, the

availability of both light and nutrients, and zooplankton grazing drive the formation and location of the SCML in meromictic lakes (Klausmeier and Litchman, 2001). Primary productivity is often highest within the SCML (Camacho, 2006).

Meromixis permits the development of laminated or varved sediments (Anderson et al., 1985). Varved sediments can record climatic transitions, vegetation changes, changes in sediment transport, and atmospheric deposition patterns (O'Sullivan, 1983). Laminated sediments are also useful for studying the formation, deposition, and diagenetic transformation

of chemically precipitated minerals (Ledesma et al., 2023). Such sediments are useful to interpret the origin of minerals in sediments deposited from past stratified marine waters when such conditions no longer exist (Swanner et al., 2020). Meromictic lakes are present on all continents but are thought to be rare (Hall and Northcote, 2012; Stewart et al., 2009). This results in changing sedimentation patterns compared to lakes that mix regularly. A sediment indicator of an anoxic monimolimnion typical of low sulfur meromictic lakes is enhanced burial of oxygen-sensitive mineral forming elements, particularly iron and



manganese (Piper and Dean, 2002). Such minerals can potentially be used as proxies for paleoredox conditions, which have application for paleo marine and lacustrine studies (Swanner et al., 2020).

Four lakes in Itasca State Park, Minnesota, USA - Arco, Budd, Deming, and Josephine (Figure 1) - have been described as meromictic in the scientific literature since the 1960s (Baker and Brook, 1971; Anderson et al., 1985; Stewart et al., 2009). However, there has never been physical and chemical data from all lakes during all seasons - especially spring and fall when mixing would be expected - to document through the maintenance of a chemocline that they are meromictic. The goals of this study are to 1) determine whether these four lakes are meromictic and the factors that drive stratification, 2) investigate the water type, sources, and reasons for meromixis, and 3) describe the unique biological features of these lakes. To achieve these goals, we conducted fieldwork from 2006-2009 and 2019-2022, and integrated data from University of Minnesota students working at the Itasca Biological Station and Laboratories since the 1950s. The description of the lakes, their mixing status, and the factors that control it provide an expanded understanding why lakes resist mixing. The identification of meromictic lakes is important as they are critical analogues for understanding of the biogeochemistry of past oxygen-stratified oceans (Swanner et al., 2020) and alterations to global biogeochemical cycles that may result from climate change and anthropogenic impacts strengthening stratification in lakes.

## 2 Materials and Methods

Anoxia and hypoxia are common features of lakes (Nürnberg, 1995), and meromixis is an end-member of lake mixing regimes in which a lake does not mix seasonally, resulting in a permanently anoxic monimolimnion. Seasonally persistent anoxic conditions, and even meromixis, are becoming more common as a result of climate change (Ficker et al., 2017), land-use changes (Jenny et al., 2016), and road salt contamination (Koretsky et al., 2012). Meromictic lakes are characterized by a sharp increase in electrical conductivity at a chemocline separating a cold and/or dense monimolimnion from the epilimnion (Boehrer and Schultze, 2008). The epilimnion is in contact with the atmosphere, while the monimolimnion is anoxic. These conditions restrict the habitat of aerobic organisms, including zooplankton and fish, and can amplify microbial or geochemical cycling of elements across the redoxcline (Busigny et al., 2016) and impact the emission of greenhouse gases such as methane (Lambrecht et al., 2020).

The stability of a lake against mixing is conferred by density differences between the epilimnion and monimolomnion, driven by the thermocline and chemocline. Lake morphometry is a key factor in the development of meromixis and can be useful to evaluate the potential for a lake to develop meromixis. Lakes that are relatively deep compared to their surface area, which determines the fetch upon which wind can act, are less likely to overturn. The potential for water column stratification is reflected by relative depth ($Z_r$; %), which is the ratio of the maximum lake depth ($Z_m$; m) to the diameter of a circle of area equal to that of the lake ($A_0$; m$^2$), expressed as a percentage.

$$Z_r = \frac{Z_m}{2} \sqrt{\frac{\pi}{A_0}} \times 100 \tag{1}$$



Most temperate zone lakes prone to meromixis have relative depths exceeding 4% and $A^0$ of less than 500,000 m² (Swanner et al., 2020). Density differences leading to the development of meromixis can also result from increased chemical constituents that enrich the water density of monimolimnion, such as from runoff, groundwater inputs, and remineralization processes associated with organic carbon loading (Hakala, 2004). Thermal stratification is typically not the main mechanism stabilizing

meromixis in lakes of the temperate zone (Boehrer et al., 2017).

Gradients of dissolved oxygen and inorganic nutrients are mediated by primary productivity and uptake in the epilimnion and sinking and decomposition of biomass in the monimolimnion (Camacho, 2006). Stratification limits the return of nutrients to the epilimnion. A subsurface chlorophyll maximum layer (SCML) often develops within the metalimnion, the middle seasonally mixed layer of meromictic lakes (Baker and Brook, 1971; Camacho, 2006). Stable stratification, the

availability of both light and nutrients, and zooplankton grazing drive the formation and location of the SCML in meromictic lakes (Klausmeier and Litchman, 2001). Primary productivity is often highest within the SCML (Camacho, 2006).

Meromixis permits the development of laminated or varved sediments (Anderson et al., 1985). Varved sediments can record climatic transitions, vegetation changes, changes in sediment transport, and atmospheric deposition patterns (O'Sullivan, 1983). Laminated sediments are also useful for studying the formation, deposition, and diagenetic transformation

of chemically precipitated minerals (Ledesma et al., 2023). Such sediments are useful to interpret the origin of minerals in sediments deposited from past stratified marine waters when such conditions no longer exist (Swanner et al., 2020). Meromictic lakes are present on all continents but are thought to be rare (Hall and Northcote, 2012; Stewart et al., 2009). This results in changing sedimentation patterns compared to lakes that mix regularly. A sediment indicator of an anoxic monimolimnion typical of low sulfur meromictic lakes is enhanced burial of oxygen-sensitive mineral forming elements, particularly iron and

manganese (Piper and Dean, 2002). Such minerals can potentially be used as proxies for paleoredox conditions, which have application for paleo marine and lacustrine studies (Swanner et al., 2020).

Four lakes in Itasca State Park, Minnesota, USA - Arco, Budd, Deming, and Josephine (Figure 1) - have been described as meromictic in the scientific literature since the 1960s (Baker and Brook, 1971; Anderson et al., 1985; Stewart et al., 2009). However, there has never been physical and chemical data from all lakes during all seasons - especially spring and

fall when mixing would be expected - to document through the maintenance of a chemocline that they are meromictic. The goals of this study are to 1) determine whether these four lakes are meromictic, 2) investigate the water type, sources, and reasons for meromixis, and 3) describe the unique biological features of these lakes. To achieve these goals, we conducted fieldwork from 2006-2009 and 2019-2022, and integrated data from University of Minnesota students working at the Itasca Biological Station and Laboratories since the 1950s. The description of the lakes, their mixing status, and the factors that

control it provide an expanded understanding why lakes resist mixing. The identification of meromictic lakes is important as they are critical analogs for understanding of the biogeochemistry of past oxygen-stratified oceans (Swanner et al., 2020) and alterations to global biogeochemical cycles that may result from climate change increasing stratification in lakes.



## 3 Results and Discussion

### 3.1 Morphometry and Mixing Status

All four study lakes exceed a $Z_r$ of 4% and have surface areas of less than 500,000 m$^2$, typical of meromictic lakes in the temperate zone (Swanner et al., 2020). Deming Lake is the largest and deepest of the four study lakes with a 54,325 m$^2$ surface area and a maximum depth of 20.8 meters (Table 1), with 4.3% of the surface area encompassing water depths at or below 17 m. The maximum depth of Deming Lake has previously been reported as 16.5 m (Hooper, 1951), 17 m (Baker and

Brook, 1971), and 17.6 m (Lascu and Plank, 2013). Josephine Lake has the longest fetch and lowest $Z_r$ of all study lakes. The maximum depth of Josephine Lake (14.8 m) has previously been reported as 10.3 m (Baker and Brook, 1971), 12-13 m (Callis et al., 1976), and 13 m (Gage and Gorham, 1985). The surface area with depths at or below 13 m in Josephine Lake is 5.1%. Budd Lake has the highest $Z_r$ (8.8%). There was the greatest variation between the previously reported maximum depth of Budd Lake (10.8 m; Baker and Brook, 1971) and measurements presented here (16.1 m). Arco Lake has the smallest surface

area of 24,180 m$^2$ and a maximum depth of 12.8 m. Arco and Budd Lakes have steep banks along their eastern sides (Figure 1).

In all lakes, steep-sided deep holes were detected that were deeper than previously reported (Figure 2; Supplementary Figure 1). Differences in technology and the small surface areas encompassing the deepest parts of the basins could account for the variation – these basins may have been difficult to locate and map with manual techniques. Alternately, the temperature,

pressure, and/or salinity gradients present at the tops of these holes could deflect sound beams, causing errors in our depth measurements (Boehrer and Schultze, 2008). Maximum depths should be verified with measurements by multiple techniques in the future.

Deming Lake has the longest record of seasonal profiles of temperature and specific conductance (2006-2009 and 2019-2022), as its sediments have previously been used to investigate Holocene climate variations (Lascu et al., 2012;

McLauchlan et al., 2013). A chemocline, or sharp increase in specific conductance, persists in Deming Lake at all sampled times (Figure 3), which is expected for a meromictic lake. The chemocline occurs at 11-13 m (Figure 3). The chemocline was below 10 m in 1989 (Church et al., 1989), and from 11-13 m a decade later (Reiter et al., 1998) based on temperature-compensated conductance measurements from IBSL student reports, indicating that the chemocline has been at a consistent depth for decades.

A systematic increase in the magnitude of specific conductance readings below the chemocline was observed from the 2006-2009 to the 2019-2022 datasets (Figure 3). This could result from variations in the amount of in-lake primary productivity over time (Campbell, 1977), or could increase with time since a mixing event (Katsev et al., 2010). Deming Lake mixed after a beaver dam broke on the western side of the lake in 1997 (Frane and Walberg, 1997). If the lake has not mixed since aside from this catastrophic event, the specific conductance should have increased from 1997 to 2006-2009 and from

2006-2009 to 2019-2022 at similar rates. The 1997 conductivity data was converted to specific conductance using the reported temperature values (Frane and Walberg, 1997), and the value at 14 m in July 1997 was compared to the values at 14 m in July



2006 and July 2021. The specific conductance values have increased between both intervals, but the rate has changed from 31 μSiemens cm$^{-1}$ yr$^{-1}$ from July 1997 to July 2006 to 11 μSiemens cm$^{-1}$ yr$^{-1}$ from July 2006 to July 2021. This could imply a partial mixing event in the years between 2006 and 2021, or some change to the supply of water and dissolved ions to the lake
between the two time intervals being compared.

Changes in precipitation could have affected the rate of the increase of specific conductance in Deming Lake over this period. Historical drought records are available for the region encompassing Itasca State Park from the year 2000 (National Integrated Drought Information System; Supplementary Figure 2). Drought over this period could have modulated the rate of the lake's specific conductance increase through increased evaporation (Jellison and Melack, 1993). Alternatively, or in
addition, the magnitude of water sources of differing specific conductance supplying the lake may have been altered (Ludlam and Duval, 2001). In the second case, drought would decrease the magnitude of low specific conductance precipitation and increase the proportion of higher specific conductance groundwater to the annual water budget. As Deming Lake does not have a surface inlet, changes in streamflow can be neglected. The amount of recharge to the groundwater system may have also decreased due to drought, effectively decreasing groundwater inputs to the lake, and counteracting the effects of
evaporation on specific conductance in Deming Lake.

The record of specific conductance in the other three lakes is only available for the period of 2019-2022. Budd Lake had the lowest range of specific conductivity values (23-90 μS cm$^{-1}$). A weakly demarcated chemocline was observed around 4-5.5 m water depths for May 2021, July 2021, October 2021, and May 2022. The January 2021 profile shows a deeper chemocline (Figure 3). Specific conductance values were uniform down to 8 m in Josephine Lake during the current study
period (Figure 3). A chemocline was only detected below 9 m in October 2021, while depths below 11 m were not assessed on other dates during the study period due to the difficulty of finding the deepest area before mapping in 2022. Autumnal circulation down to 10 m was reported in Josephine Lake in 1975 (Gage and Gorham, 1985). A chemocline was also present at all time points for Arco Lake between 7-10 m except for May 2022 (Figure 3). It was poorly developed in May 2021, and its deepest occurrence was in January 2021. Arco Lake has a maximum depth of 12.6 m, but no specific conductance data was
collected below 12 m as part of the profiles acquired during the study period due to the difficulty of finding the small deep spot (representing 1.7% of the surface area). The chemocline is shallowest in mid-summer, consistent with the 7 m chemocline observed in temperature-equilibrated conductivity readings in 1975 and 1976 (Evans and Bjerklie, 1975; Barnes et al., 1976), and 2011 (Harren et al., 2011).

Data from the temperature sensors (1.5, 3.5, 5.5, 7.5, 9.1, 9.5 m) and a conductance sensor (9.1 m) in Arco Lake from
May 2021 to May 2022 are presented in Figure 4. In the summer of 2021, Arco Lake developed a summer thermocline. The temperatures at different depths started to converge in late fall until the lake became isothermal and continued cooling from 6 to 4°C in mid-November 2021, representing the erosion of thermal stratification. The Brunt-Väisälä or buoyancy frequency (N), commonly expressed as $N^2$ in units s$^{-2}$, calculated from the time series temperature data is near zero at that time, indicating an erosion of stability, which persisted until the sensors were removed on May 16, 2022, over a week after the ice-off on May
8 (Supplementary Figure 3). A drop in the conductance value at 9.1 m in early November 2021 preceded isothermal conditions



and indicated mixing down to at least 9.1 m (Figure 4). The dimensionless lake number (Imberger and Patterson, 1989) was calculated using the time series and hourly wind data from Itasca State Park (Supplementary Figure 4). It started to consistently rise on May 15, 2022, as the epilimnion warmed. In combination with the seasonally variable chemocline depth (Figure 3), these analyses indicate that Arco Lake mixes during isothermal periods and is not a meromictic lake. The data presented is

most consistent with Arco being a dimictic lake, although holomixis cannot be ruled out.

Data from the temperature sensors (0.5, 2.5, 5.5, 8.5, 11.5, and 14.5 m) in Deming Lake from June 2019 to May 2020 are presented in Supplementary Figure 5. The lake became isothermal in early November 2019. The Brunt-Väisälä frequency calculated from this data dropped to near zero and rebounded slightly due to under-ice thermal stratification, a phenomenon not observed in Arco Lake (Supplementary Figure 3). The specific conductance profiles in Deming consistently show the same

trend, although the exact depth and magnitude of the chemocline vary seasonally and on the decadal scale (Figure 3). This indicates partial seasonal mixing in Deming, likely during isothermal periods, but it may be insufficient or of too short a duration to fully mix the lake. Deming was ice-free on April 25, 2020 (Lake Ice Out Dates, 2022), and the lake number started rising immediately due to the onset of thermal stratification (Supplementary Figure 4), limiting the opportunity for mixing. Long-term seasonal observation of a chemocline is consistent with Deming being a meromictic lake (Zadereev et al., 2017).

Arco presents an end-member of a seasonally mixed lake (dimictic or holomictic) and Deming an end-member of a meromictic lake among the four lakes studied here. In the absence of high-frequency sensor profiles for Budd and Josephine Lakes, the Schmidt stability (Supplementary Figure 6) and the Brunt-Väisälä frequency (Supplementary Figures 7-10) were calculated from seasonal profiles of temperature (Supplementary Figure 11) and salinity for all four lakes. The Schmidt stability indicates that Deming and Budd Lakes are the most strongly stratified, followed by Josephine and Arco Lakes. It is

unknown when ice came off Budd Lake in 2022, but Deming Lake was ice-free on May 7, and Arco and Josephine Lakes were ice-free on May 8 (Lake Ice Out Dates, 2022) following strong winds on the evening of May 7. Likely, Budd Lake was also ice-free by May 8. A chemocline consistent with other summer observations was present in Budd Lake on May 17, 2022, which could indicate a lack of spring mixing. However, the January 2021 specific conductance profile for Budd Lake shows a very weak and deep chemocline, which would be expected following an autumn mixing event. Without full profiles of specific

conductance in Josephine Lake, it is impossible to evaluate mixing. However, based on existing profiles, if a chemocline persists through spring turnover in Josephine Lake as suggested by the sharp and deep chemocline observed in May 2022, then the monimolimnion must be limited to depths below 11 m, representing only 3.6% of total lake volume (11,404.5 $m^3$ of 313,295.5 $m^3$).

Water colour is related to the abundance of dissolved organic carbon (DOC) (Pace and Cole, 2002). Of the four study

lakes, Budd Lake had the most coloured water, followed by Deming Lake (Supplementary Table 1). Arco and Josephine Lakes had very little colour. Visually, Budd and Deming Lakes appear brown, while Arco and Josephine Lakes appear green. Enhanced water colour could lead to a shallower thermocline and stronger stratification in Budd and Deming Lakes due to greater light absorption by compounds conferring colour (Houser, 2006). The Brunt-Väisälä frequencies, calculated from temperature and salinity profiles, were generally highest in the epilimnia during late summer (July 2021; Supplementary





Figures 7-10). This indicates that temperature is more important than salinity to the stability of these lakes. In August 2022,
when Deming Lake had the highest $N^2$ values observed from the 2019-2022 dataset (data not plotted in Supplementary Figure
9), the meromictic stability (S') (Walker, 1974) was 7.81 J m$^{-2}$, within the range observed for the meromictic and ferruginous
Lake 120 in Canada (Campbell, 1977) and Lake Nordbytjernet in Norway (Hongve, 1999). These observations suggest that
meromixis in Deming Lake is primarily due to thermal stratification, with a small contribution from salinity that limits mixing

in this lake but not the others nearby. This phenomenon could be common to dilute meromictic lakes. The mixing event at
Deming Lake in the summer of 1997 due to a breached beaver dam west (Frane and Walberg, 1997) indicates that mixing can
occur due to catastrophic events.

### 3.2 Chemical Characteristics and Water Sources

A Piper diagram was used to define the water types of the four lakes based on their major cations and anions. The

predominant cation in all four study lakes was calcium (Figure 5). Budd Lake had a greater proportion of milliequivalents from
sodium and potassium than the other three lakes, with the balance mostly from magnesium. Bicarbonate and carbonate ions
represented nearly all anion milliequivalents in Deming, Arco, and Josephine Lakes whereas chloride and sulfate contributed
to the anion balance in Budd Lake. The water type for Deming, Arco, and Josephine Lakes is a calcium-bicarbonate type,
whereas Budd Lake does not have a dominant water type. The low specific conductance and lack of water type in Budd Lake

indicate the water source is predominantly precipitation. Calcium, magnesium, and carbonate/bicarbonate ions source from
the calcareous Itasca moraine and are typical for the Itasca region (Megard et al., 1993). This indicates that water supplying
Deming, Arco, and Josephine Lakes has undergone more water-rock reaction than the water supplying Budd Lake, such as
might be expected from an increasing proportion of groundwater seepage to a lake's water inputs. Deming, Arco, and Josephine
Lakes lie in a tunnel valley channel containing coarser sands and gravel while Budd Lake is poorly integrated hydrologically

with the other three lakes, and the net seepage in Budd Lake is probably the lowest among all the lakes. Arco Lake (465.8
MAMSL) is located 51 meters north of Josephine Lake (465.4 MAMSL) and Deming Lake (464.8 MAMSL) is located 284
meters north of Arco Lake. The similar lake levels and wetlands between Arco and Deming Lakes (Figures 1 and
Supplementary Figure 1) indicate these lakes are likely to be hydrologically connected. If the water table mimics the surface
topography, groundwater potentially flows from Arco-Josephine to Deming Lake. However, Budd Lake (478.6 MAMSL) is

perched highest in the watershed, located off-axis of Josephine-Arco-Deming Lakes, to the west of a ridge (Figure 1). Because
Budd Lake is perched, the ridge is likely composed of less permeable material, possibly till, than the valley sediments. The
presence of wetlands on the north end of Budd Lake that follow a small valley towards Deming Lake suggests that there could
be a surface or near surface hydrological connection from Budd Lake to Deming Lake. Lakes highest in their watershed are
more likely to receive a greater proportion of water from precipitation (Kratz et al., 1997), consistent with the dilute nature of

Budd Lake water.

The isotopic composition of water (i.e., $\delta^2H_{H2O}$ and $\delta^{18}O_{H2O}$ in ‰) in May 2021 varied little with depth in each lake
(Supplementary Figure 12). In Arco, Budd, and Deming Lakes sampled in June 2019, the epilimnion waters were depleted



relative to the deeper water. Additional measurements at Deming Lake in July and October 2021 show enriched values in the epilimnion relative to the deeper water. Preferential evaporation of light isotopes during the summer and early fall drives the
epilimnia to more enriched values. In Figure 6, all lakes data points diverge from a local meteoric water line (LMWL; Stelling et al., 2021), falling on a lake evaporation line (LEL) with equation $\delta^2H_{H2O}$ (‰) = 4.5*$\delta^{18}O_{H2O}$ (‰) - 29.9. The intersection of the LMWL and the LEL is the composition of regional groundwater, representing isotopically depleted snowmelt as the predominant source of recharge to the local aquifers (Krabbenhoft et al., 1994).

To ascertain the isotopic composition of groundwater in the area, historical $\delta^2H_{H2O}$ and $\delta^{18}O_{H2O}$ data from springs at
Elk Lake in Itasca State Park were retrieved from the Minnesota Spring Inventory (Minnesota Department of Natural Resources, 2022). The Elk Lake springs were sampled again in May 2022, and samples were collected from spring along Nicollet Creek and the bog on the southeast side of Deming Lake (Supplementary Table 2). These sites had abundant marsh marigolds (Supplementary Figure 13), which bloom in May at groundwater discharge sites in northern Minnesota (Rosenberry et al., 2000). The sites at Elk Lake and Nicollet Creek had visible iron mineralization, indicating reducing, iron-bearing water
discharge at the surface (Supplementary Figure 13). The Nicollet Creek spring sample lies closest to the intersection of the LEL and LMWL, whereas the Deming bog sample lies closest to the lakes but is more enriched than the lakes (Figure 6).

The seasonal amplitude of a lake's oxygen isotopes can be used to calculate a lake's water residence time (Engel and Magner, 2019). A time series of temperature data from weather station ITCM5 was used to determine average daily temperatures, and seasonal maximum and minimum temperature values were used to calculate the amplitude in $\delta^{18}O_{H2O}$ inputs
of precipitation and water vapor $\delta^{18}O_{H2O}$ (Supplementary Figure 14). These two values were used to estimate a mean water residence time of 100 days for Deming Lake. The Deming Lake $\delta^{18}O_{H2O}$ values encompass early spring, mid-summer, and late summer data, but data from the other three lakes was only from May and June time points (Supplementary Figure 12), and so the mean water residence time could not be calculated by this method.

Potential water sources to Deming Lake are precipitation, outflow from the boggy areas to the south and west, and
groundwater. In addition to higher specific conductance values than the other lakes, Deming also has higher concentrations of iron in the monimolimnion than the other lakes, exceeding 1 mM, and has sediments enriched in biogenic magnetite (Lascu et al., 2012). As groundwater in Itasca State Park has visible iron mineralization (Supplementary Figure 13), it is possible that groundwater supplies iron into Deming Lake (Swanner et al., 2020). Due to active iron redox cycling, which scavenges iron into the monimolimnion in meromictic lakes, dissolved iron can accumulate to high concentrations in its ferrous form (Busigny
et al., 2016; Campbell, 1977).

### 3.3 Biological Characteristics

Prior work on these four lakes indicated the presence of a lake-wide turbidity maximum layer that was persistently just below 5 m in Deming Lake (Baker and Brook, 1971). This peak was observed between late May and August from 1968-1970 but was absent in the winter. It was reported in that study that bacteria and phytoplankton both contributed to the turbidity





maximum based on microscopic observations. This work indicated the layer was populated by the cyanobacteria *Oscillatoria agardhii* var. *isothrix* (this genus is now called *Planktothrix*).

From 2019 to 2022, chlorophyll-a measured via multi-wavelength fluorescence acquired with a Fluoroprobe allowed for deconvolution of the fluorescence signal so chlorophyll-a could be attributed to one of four taxonomic groups: Cyanobacteria, Chlorophyta, Diatoms and Dinophyta, or Cryptophyta. Taxon-specific concentrations of chlorophyll-a in each

lake inform phytoplankton community structures with depth (Figure 7). Deming Lake has a persistent SCML around 5-6 m attributable to Cyanobacteria. The SCML was less distinct in May 2022, one and a half weeks after ice off. It was also less distinct in May 2021. This suggests that the SCML in Deming Lake is a summer phenomenon that develops at or below the thermocline (Supplementary Figure 11), consistent with the observations from Baker and Brook (1971). In June 2019, a profile of photosynthetically active radiation was also recorded (Supplementary Figure 15), and showed an inflection point at 5.5 m,

corresponding to the depth of the SCML at that time. These observations are consistent with cyanobacteria forming a dense accumulation at that depth, attenuating the light flux into deeper waters.

Samples from the epilimnia or the SCML were taken from the four lakes in May 2022: Arco (5 m), Budd (4.5 m), Deming (3.5 m), and Josephine (4.5 m). The most common phytoplankton of the four lakes were the cyanobacteria *Planktothrix* sp. and *Aphanocapsa* sp., and the green alga *Monoraphidium* sp. (Supplementary Figure 16). Some other phytoplankton

species identified were the red alga *Cryptomonas* sp., the cyanobacteria *Planktolyngbya* sp. and *Dolichospermum* sp., as well as other colonial green algae and cyanobacteria species.

All four lakes have subsurface oxygen maxima exceeding air saturation during the summer but lack this feature in the fall (Figure 8). The subsurface oxygen maximum in Deming Lake usually occurs at 4 m, near the top of the thermocline (Bieter et al., 1991; Church et al., 1989; Balk et al., 2007). Metalimnetic oxygen maxima can result because dissolved oxygen

is more soluble in cold water after spring mixing, but as the water warms the gas solubility decreases, causing oxygen to become supersaturated. Stratification induced by the thermocline then limits the discharge of supersaturated oxygen to the atmosphere (Wilkinson et al., 2015). Deming Lake rapidly develops a thermocline after ice-off (Supplementary Figure 6), providing a mechanism for gas trapping. Enhanced biological productivity can also contribute to the oxygen maximum, but is generally a smaller contributor (Wilkinson et al., 2015; Craig et al., 1992). The subsurface oxygen maximum is above the

SCML in Deming Lake, suggesting that the entrainment of spring oxygen is a larger contributor than photoautotrophy to the oxygen maximum. However, there is also a pH maximum at that 4 m during the summer (Bieter et al., 1991; Supplementary Figure 18; Church et al., 1989; Barkow and Habedank, 1990), as is expected for oxygenic photosynthesis. Itasca Biological Station and Laboratories student experiments that quantified photosynthesis and respiration via dissolved oxygen measurements in bottle experiments generally found that net oxygen release was minimal in the SCML due to vigorous

respiration (Barkow and Habedank, 1990; Engstrom et al., 1974). Measurements of the $O_2$/Ar ratio would help to quantify the contribution of oxygen trapping vs. oxygenic photosynthesis to the Deming Lake subsurface oxygen maximum (Craig et al., 1992).



The occurrence of a summer SCML in Arco, Budd, and Josephine Lakes was more variable than in Deming Lake (Figure 7). Arco Lake had a spring bloom of cyanobacteria in May 2021 and in May 2022, at which time the highest chlorophyll-a of this study was recorded. At both times, the SCML occurred below the oxygen maximum, in a hypoxic zone of the water column. The May 2022 SCML at 5 m in Arco Lake also enhanced light attenuation below that depth (Supplementary Figure 15), as was observed in Deming Lake in June 2019.

Arco, Budd, and Josephine Lakes also have subsurface oxygen maxima during summer, although not as consistently as Deming. When the depth of the thermocline is plotted against the depth of the oxygen maximum for all four lakes, there is a significant correlation if October 2021 samples are excluded (Supplementary Figure 19). In the fall, the subsurface oxygen maximum is absent (the highest values are at the surface), and the thermocline is deepest (Figure 8). This may reflect that in fall the thermocline has deepened into hypoxic or anoxic waters.

## 4 Conclusions

Although reported to be meromictic, based on seasonal temperature and specific conductance profiles, Arco, Budd, and Josephine Lakes may be holomictic or even dimictic. Deming Lake has been meromictic in the periods of data collection presented here (2006-2009 and 2019-2022) but experienced a mixing event in 1997 when an adjacent beaver dam broke. Although a seasonally persistent chemocline occurs in Deming Lake, stratification is predominantly a thermal phenomenon, with stability mostly conferred by a thermocline that develops rapidly after isothermal spring and fall periods. Thermal stratification is not typical of meromictic lakes in the temperate zone, but this weak stratification may be common in dilute boreal lakes (Meriläinen, 1970; Campbell, 1977). However, such meromixis may be easily perturbed by hydrographic or land-use changes (Hongve, 2002), which is consistent with Deming Lake mixing in 1997.

The chemical composition of the study lakes reflects their position in the watershed and water sources. The highest-elevation lake, Budd Lake, had no distinguishable water chemical type and likely received most of its water from precipitation. Arco, Deming, and Josephine Lakes are all Ca-Mg-bicarbonate type waters (Figure 5). Deming Lake had the highest ionic load, consistent with a greater contribution of water that had undergone some water-rock interaction, likely groundwater. Further work is necessary to characterize groundwater composition and quantify and localize groundwater input to determine if Deming Lake could be classified as crenogenic meromictic. The short water residence time in Deming Lake implied by the analysis of $\delta^2H_{H2O}$ and $\delta^{18}O_{H2O}$, along with the minimal surface water inputs and likely high permeability of tunnel valley deposits suggest the shallow groundwater system should be considered as a major water source.

All four lakes have SCML during summer. The depth of the SCML varies in Arco, Budd, and Josephine and is sometimes absent, but persists at around 5 m in Deming, corresponding to the base of the photic zone. Subsurface oxygen maxima above air saturation likely develop by trapping gas in cold water that then warms with rapid thermocline development in the spring. Corresponding pH maxima in Arco and Deming indicate that oxygenic photosynthesis could also contribute to the subsurface oxygen maxima.




**Code availability** Upon acceptance, data analysis codes will be made available through GitHub.

**Data availability** Water properties of Arco Lake, Budd Lake, Deming Lake, and Josephine Lake in Itasca State Park from 2006-2009 and 2019-2021, v. 2. Environmental Data

Initiative. https://doi.org/10.6073/pasta/6f4292da3adcc029934805fdb9a314d4

**Supplement link**: the link to the supplement will be included by Copernicus, if applicable.

**Author contribution**: Individuals in parenthesis contributed substantially to the study's conception (EDS), data acquisition (EDS, CH, SAA, IL, GL, PP, HS, SD, KB, AB, LB-T, TC, OJ, LN), or analysis (EDS, CH, SAA, GL, PP, HS, SD, KB, AB, LB-T, TC, OJ, LN); contributed substantially to drafting the manuscript (EDS, CH, SAA, GL, PP, HS, SD, KB, AB, LB-T,

TC, OJ, LN); and approved the final submitted manuscript (EDS, CH, SAA, IL, GL, PP, HS, SD, KB, AB, LB-T, TC, OJ, LN).

**Competing interests**: The authors declare that they have no conflict of interest.

**Acknowledgements:** This work was supported by a National Science Foundation CAREER award to EDS (1944946). The GeoAllies program, funded by the National Science Foundation, supported student participation in the GEOL 406/506

field course, as did funds from the Department of Geological & Atmospheric Sciences at Iowa State University. Support for the field course was also provided by Dr. Jonathan Schilling, Director of the Itasca Biological Station and Laboratories, and Dr. Emily Schilling. Phytoplankton was identified at the genus level with the help of Dr. Lesley Knoll, Associate Director of the Itasca Biological Station and Laboratories, who also provided assistance with field operations for the duration of the 2019-2022 field seasons. Chad Wittkop and Moses Lado (Minnesota State University) assisted with fieldwork in 2019. Jazlyn Beeck

and Elise Bauer (Iowa State University) assisted with fieldwork in 2021. Julia Kelly and Shannon Farrell of the University of Minnesota Library assisted with accessing Itasca Biological Station and Laboratories student reports.

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







**Figure 1: Map of the four study lakes within Itasca State Park, Minnesota. Basemaps: USGS, ESRI.**





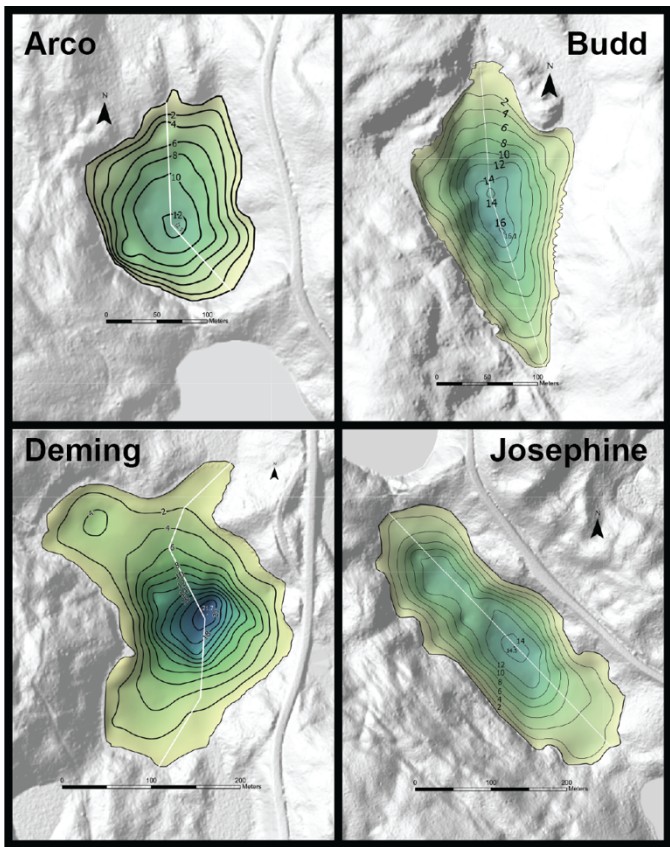

**Figure 2. Bathymetric maps of the study lakes overlain on the digital elevation models. Depth is in meters. The fence diagrams correspond to the cross-sections in Supplementary Figure 1.**





**Figure 3. Specific conductance profiles of a) Arco Lake, b) Budd Lake, c) Josephine Lake, and d) Deming Lake. For**
**Deming Lake, seasons were classified according to solstice and equinox dates.**





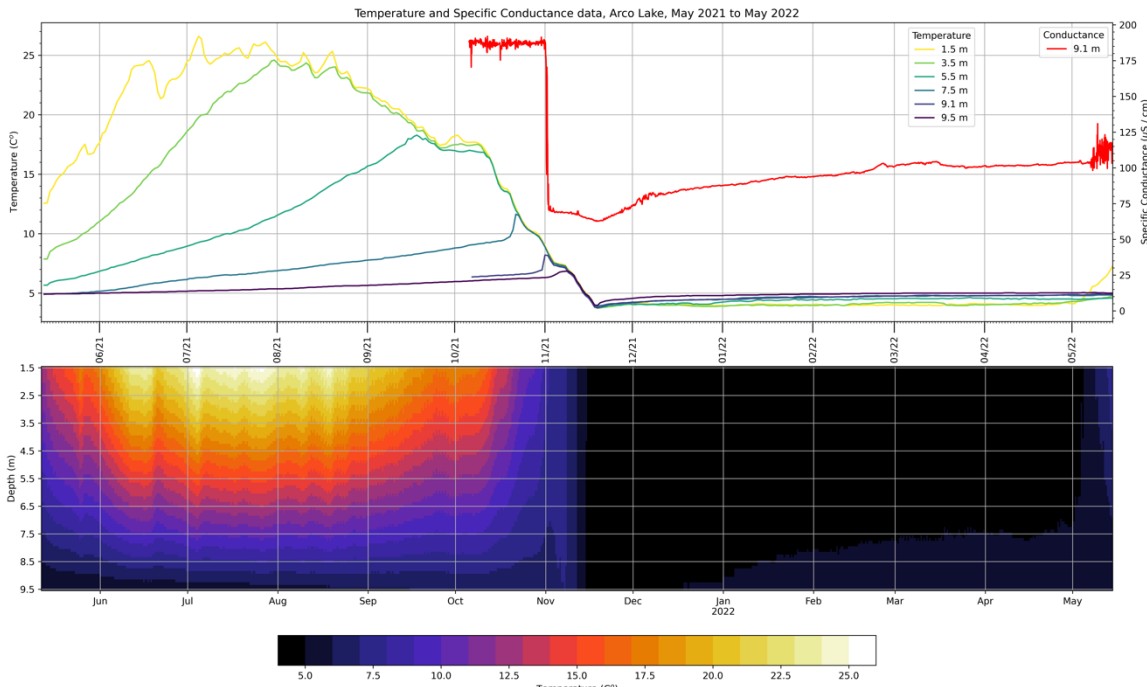

**Figure 4. Hobo sensor data for temperature and conductance from Arco Lake, collected from May 2021 to May 2022 and from October 2021 to May 2022, respectively (top). Isotherm plot of the temperature data from the Hobo sensor (bottom).**




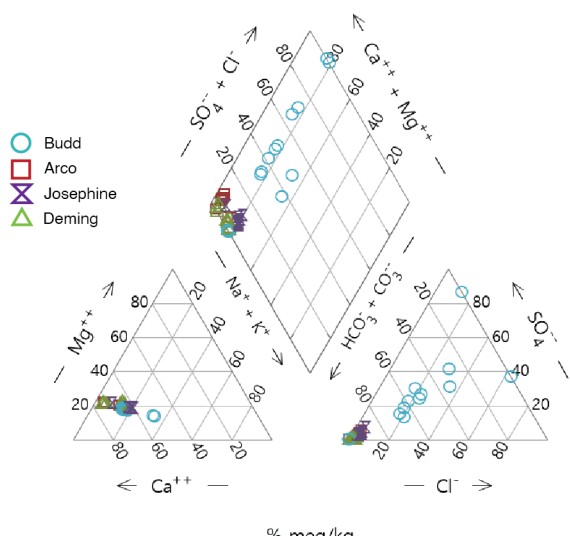

**Figure 5. Piper Diagram of major cations and anions collected in May 2021.**

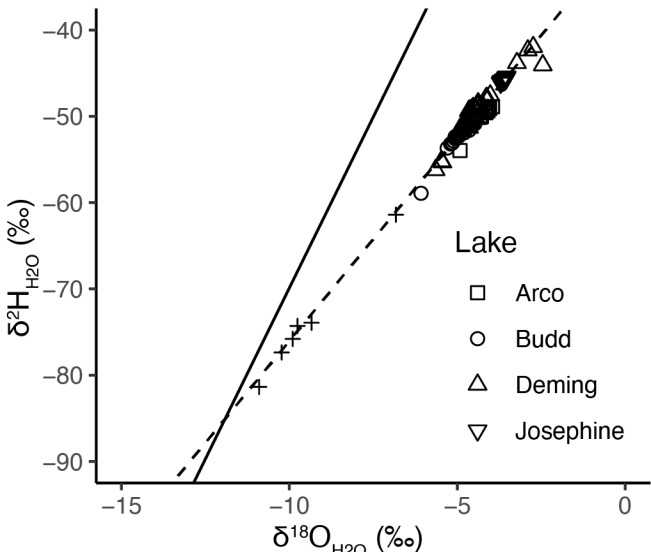

**Figure 6. Isotope data and fit (dashed line) of an evaporation line ($\delta^2H_{H2O}$ = 4.5*$\delta^{18}O_{H2O}$ - 29.9, r-squared = 0.93) from the four study lakes plotted with a Local Meteoric Water Line (solid; Stelling et al. 2021). Crosses are spring or bog water data from Itasca State Park (Supplementary Table 1). The cross closest to the lake data points is the Deming bog. Analytical precision is within the symbol sizes.**



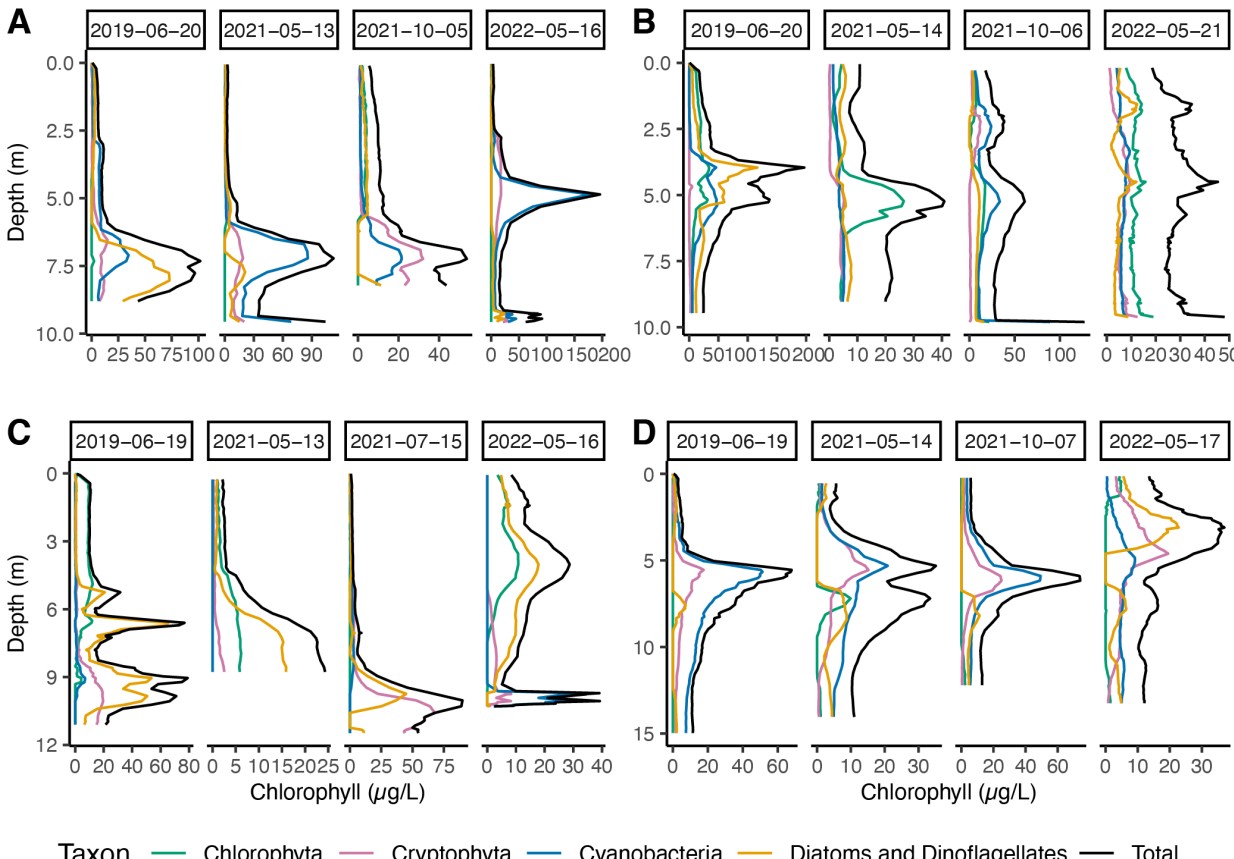

Figure 7. Multi-wavelength chlorophyll fluorescence (i.e., Fluoroprobe) measurements at a) Arco, b) Budd, c) Josephine, and d) Deming Lakes, showing seasonal and depth trends in major taxonomic groups. Note that the x scales vary between plots.




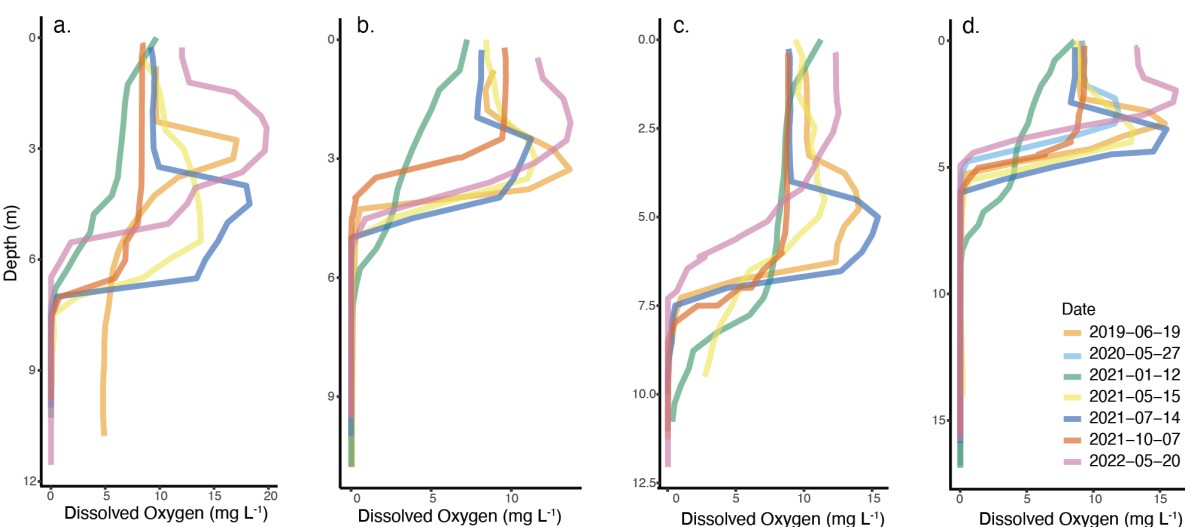

**Figure 8. Dissolved oxygen measurements at a) Arco, b) Budd, c) Josephine, and d) Deming Lakes, showing seasonal and depth trends.**

**Table 1. Morphometric parameters for the four study lakes.**

| Lake | Volume ($m^3$) | Surface Area ($m^2$) | Maximum Depth (m) | Relative Depth (%) | Fetch Distance (m) | Fetch Heading (degrees) |
|------|----------------|----------------------|-------------------|--------------------|--------------------|-------------------------|
| Arco | 142,628.73 $m^3$ | 24180 $m^2$ | 12.6 m | 7.3% | 210.4 m | 175 deg. |
| Budd | 183,441.22 $m^3$ | 28184.5 $m^2$ | 16.1 m | 8.8% | 310.3 m | 169 deg. |
| Deming | 242,701.12 $m^3$ | 54325 $m^2$ | 20.8 m | 7.3% | 346.7 m | 188 deg. |
| Josephine | 315,765.05 $m^3$ | 52711 $m^2$ | 14.5 m | 5.7% | 432.4 m | 136 deg. |