# Peer review of "Thermal stratification and meromixis in four dilute temperate zone lakes"

_EGUsphere, 2023_

## Author Response (AR1)

IOWA STATE UNIVERSITY
OF SCIENCE AND TECHNOLOGY

Department of Geological and
Atmospheric Sciences
253 Science I
2237 Osborn Drive
Ames, Iowa 50011-3212
515 294-4477
FAX 515 294-6049

January 11, 2024

*Biogeosciences*

**Re: Manuscript from Swanner et al.**

Dear Dr. De Jonge,

I am resubmitting a revised version of egusphere-2023-1764. "Thermal stratification and meromixis in four dilute temperate zone lakes". I want to first thank you for your efforts. I see how much you had to search for reviewers, and I am very grateful that you persevered. This was a fun project that involved many students and I think will help to set a foundation for future biogeochemical investigations of this lake.

I have addressed all the reviewer comments in the revision and provided point-by-point responses amended here, as well as a marked-up version of the file changes. Notably, the methods are now included, which addresses most of the technical concerns with the manuscript. Most of the other comments fell into the category of clarifying the messaging. I believe I have addressed this with a reorganization of the introduction and movement of some context from introduction into the results and discussion and adding additional clarifying points and clearer conclusions throughout.

Thank you for the opportunity to revise the manuscript and I hope that the revised version will be acceptable for publication in *Biogeosciences.* Please let me know if you have additional suggestions to improve the manuscript.

Sincerely,

*E. Swanner*

Dr. Elizabeth D. Swanner
Associate Professor
Department of Geological & Atmospheric Sciences
Iowa State University

**RC1**

This manuscript investigates the thermal stratification and meromixis of four adjacent lakes (Arco, Budd, Deming, and Josephine) within the Itasca State Park in Minnesota, USA. Its relevance and main motivation are to evaluate if the reported lakes can be classified as meromictic. The authors want to assess if these lakes remain stratified through seasonal mixing events, and what factors contribute to their stability. Results show that all Arco, Budd, and Josephine Lakes may be holomictic or even dimictic and that Deming Lake is likely meromictic. I think this study will make a relevant contribution to the field of limnology. I recommend the publication of this manuscript after the following comments are addressed.

**Major comments:**

It was not easy to follow the manuscript results because the Materials and Methods section is very incomplete. Additionally, after reading the manuscript. I think that it would be relevant to understand what type of monitoring procedure would ensure a more solid conclusion regarding the lake's classification. I think that this aspect of the study needs to be discussed.

*I added text to the discussion/conclusion about best practices for monitoring procedures to determine mixing classifications.*

**Specific comments:**

Line 39-50: "The stability of a lake against mixing is conferred by density differences between… meromixis in lakes of the temperate zone (Boehrer et al., 2017)." Why do you have this text in two different locations?

Line 120- " The goals of this study are to 1) determine whether these four lakes are meromictic, 2) investigate the water type, sources, …to global biogeochemical cycles that may result from climate change increasing stratification in lakes." The same with this text.  Why do you have this text in two different locations?  This text belongs to the introduction. In fact, the majority of the text that is included in the Material and Methods section belongs to the introduction (I´m not saying that you must include all of text in the introduction section). In this section (Material and Methods) you must describe for example the lake's location, the location of the sampling points (profiles of temperature and specific conductance; temperature sensor chains) and the mathematical concepts considered in the analysis (e.g. Brunt-Vaisala or buoyancy frequency (N) equation; The dimensionless lake number equation). This section should also identify all the dataset's sources. For example, you only mention the water colour datasets in the results section. In my opinion this section needs to be completely reformulated.

*Regarding the two above points, it appears that when I transferred the manuscript to the template provided by the journal, I copied the Introduction twice – once into section 1 and the second time into section 2. The Materials and Methods are missing. They are now included. I apologize for this oversight.*

Line 150: Can you include the lakes sampling points location in Figure 3)?

*We did not sample from a fixed mooring. The methods now include this statement, "Measurements were made and samples were collected from a boat anchored within the deepest basin of each lake."*

Line 212: (Supplementary Figures 7-10). I suggest considering the same scale range in all figures.

*Adjusting the scale to each dataset allows the trends to be visibly resolvable, and makes for easy visual comparison of one time point to another. If I kept the scales the same I would need to change the number of plots in the paper width, so it would be harder to compare lake to lake. As a compromise, I have added the statement, "Note the variability in the x and y axis scales" to the figure legend.*

Line 251 Please replace MAMSL with: Meters above mean sea level (MAMSL). This is the first time the acronym appears in the text.

*This acronym is defined in the second paragraph of the methods.*

Line 253 – I think you mean (Figures 1 and Supplementary Figure 2).

*Supplementary Figure 2 is the drought record. Supplementary Figure 1 contains a cross-sectional lake profile showing the lake level referred to in this figure, and corresponds to the cross-sections identified in Figure 1.*

Line 212 - Figure 6. Caption. Crosses are spring or bog water data from Itasca State Park (Supplementary Table 1) This caption is correct? Table 1 shows water color in mgPt L-1

*Thanks for catching this. It has been changed to Supplementary Table 2.*

Line 275: "The Nicollet Creek spring sample lies closest to the intersection of the LEL and LMWL, whereas the Deming bog sample lies closest to the lakes but is more enriched than the lakes (Figure 6)." I suggest including these samples in Figure 6.

*The last sentence of the caption has been modified to clarify, "The cross closest to the lake data points is the Deming bog sample, and the cross closest to the intersection of the LEL and LMWL is the Nicollet Creek sample."*

Line 317. "Deming Lake rapidly develops a thermocline after ice-off (Supplementary Figure 6)". I think that the figure number is not correct.

*This should be Supplementary Figure 5.*

Supplementary Figure 19. Can you please describe the meaning of the gray area?

*This is the 95% confidence interval. I have added this information to the caption.*

**RC2**

Summary:

This manuscript attempts to characterize the mixing regimes of three lakes in Minnesota, USA. Described as meromictic within previous literature, this paper aims to assess whether this classification is accurate using field data. In addition, the paper describes some of the chemical and biological characteristics of the study lakes.

General comments:

- More needs to be included to justify the work. What is the novelty of the study as well as the relevance to the wider field. At the moment, I don't feel this is included in either the abstract, introduction, or discussion.

*Line 75 from introduction addresses this point: "The identification of meromictic lakes is important as they are critical analogues for understanding of the biogeochemistry of past oxygen-stratified oceans (Swanner et al., 2020) and alterations to global biogeochemical cycles that may result from climate change and anthropogenic impacts strengthening stratification in lakes."*

*However, your point is taken. I have completely rearranged the introduction and removed some extraneous information and added additional statements to clarify the relevance and justify the work. Additional statements were also added into the abstract, discussion, and conclusion.*

The methods are incomplete and organization needs to be improved. See my specific comments on what is missing.

*When copying the manuscript to the Biogeosciences template, I mistakenly copied the introduction twice instead of the methods. I apologize for this mistake, but the editor recommended this instead of re-uploading a corrected version. I believe the methods will address many of your comments.*

- Why do we care about the biology/chemistry and how does it contribute to the characterization of the mixing regimes and/or the impact of that mixing. Need to link together the physics and the biology/chemistry to make a more cohesive story and

*This point is now addressed throughout the revised version.*

- Conclusions on the mixing regimes of some of the lakes is drawn from a small sample (7 profiles) and incomplete data collection (not to the maximum depth of Arco). Some

acknowledgment, at the least, needs to be included that discusses how the data availability impacts the conclusions that are being drawn.

*We acknowledge there is a data limitation with the seasonal monitoring approach and even with the deployment of sensors in these small lakes. This is addressed in the discussion and with clarifying sentences suggested by RC1.*

Specific comments:

Introduction:

- More needs to be made to highlight the novelty of the study and/or the relevant contribution that this will make to the field in the introduction. E.g. are meromictic lakes understudied, very numerous, relevant to global processes? This occurs slightly in the final paragraph but is not sufficient.

*Please see my earlier comments and suggested reorganization of the introduction and additional statements throughout the text.*

Line 39-45 references are needed for these statements and the equation.

*This equation was taken from Wetzel 2001 textbook and is assumed to be well-known, but this reference was added.*

- Lines 50-55: SCML are not distinct for meromictic lakes so why particularly are you interested in them for this study?

*As shown by the data, the SCML in these lakes are well-documented, and of particular interest for the abundance of cyanobacteria, which has relevance for their utility in understanding past stratified systems. This point is now clearly made and referenced.*

- Overall, the introduction reads more like a review or a summary of information about meromictic lakes. You should include more text about what is not known in the field, why Is it relevant that we understand mixing regimes for these individual lakes. This is done a bit in the final paragraph but should be done throughout to build toward the research questions and integrate the information from the preceding paragraphs.

*See above comments.*

Methods:

*I believe all of the following comments are addressed in the methods, which were omitted from the uploaded template by mistake.*

- What is the accuracy/precision of the sensors?

- What depths are the sensors and water samples collected? (spatial resolution). This is reported in the results (line 189, 201) but should be in the methods. When did sampling occur (timings, frequency)? What was the duration of the sensor deployment?
- Include information on what/how/where pH, PAR, chlorophyll a and phycocyanin were measured (frequency/depths etc.)
- "Rasters and volume data have been deposited with the Environmental Data Initiative…" – and used to calculate the max/mean depths and relative depths as per equation 1? The methods to generate the information in Table 1 needs to be detailed
- "Student reports from courses taking place over several decades at the Itasca Biological Station and Laboratories (IBSL) (Knoll and Cotner 2018), formerly the Itasca Biological Station, were acquired from the library at the University of Minnesota, Twin Cities." More information is needed on these reports - how many, the timing, frequency etc. Also, the Knoll & Colner (2018) citation is not in the list of references.
- Precipitation data is also used but not detailed in methods
- Overall, the organization of the methods was difficult to follow. Use subheadings to guide the reader through the method types (e.g. manual sampling, lab methods, high-frequency sampling etc.)
- Physical metrics (schmidt stability, lake number, buoyancy frequency) calculations not in the methods.
- Spring/creek isotopic composition data collection not in the methods. Need information about these sites/locations and the rationale behind the collection.

*All of these points have been addressed.*

Results/discussion:

- Could Figures 1 and 2 be combined to have the morphometry in the landscape context. Currently the scale bars on Figure 2 are difficult to read and to notice the differences in scales. It should be noted that the scale bars differ among the individual lake maps if it is to be kept in the current format (which would be my last option)

*This was not possible due to the scales of the lakes being different and needing to make everything visible, as well as the different basemaps used in the two figures. However, the font and scale have been increased in Figure 2.*

- Differences in units on the x-axis on Figure 3 should be noted in the caption or made consistent

*We added a sentence telling the reader to note the variable x-axes.*

- Line 200: dimictic is a subset of holomixis no separate.

*Only holomixis is used now.*

- The data from high frequency sensors show similar patterns at Arco and Deming (isothermal in fall) but you make different conclusions about mixing regimes. Why?

*The specific conductance data show maintenance of a chemocline in Deming Lake (lines 150-151), while this is not the case for Arco (lines 195-196).*

- Comparisons of Schmidt stability values among lakes is not appropriate due to differences in depth/volume/surface area (lines 213-214). These need to be normalized (see Winslow et al., 2017 as an example) or another unitless metric used.

*We implemented the Schmidt stability calculation in rLakeAnalyzer package, which "was formalized by Idso (1973) to reduce the effects of lake volume on the calculation (resulting in a mixing energy requirement per unit area)." The values of Schmidt stability are per surface area in the units reported ($J/m\textasciicircum2$). Furthermore, the volumes, depths, and surface areas of the lakes are quite comparable (Table 1). We do not have high resolution buoy data for all lakes, as in the Winslow et al., 2017 study referenced that would be necessary for the temporal normalizations described in that work. We do have the thermistor data for Arco and Deming, but don't feel that is appropriate to compare with the seasonal profiles from Josephine and Arco, and thus present the Schmidt stability calculations only from the seasonal profiles in Supplementary Figure 6.*

- Line 206 reference to lake ice out dates 2022, is this the right author? Use of "Likely" needs justification, why is it likely?

*This reference refers to a state database. It is how the reference manager interpreted the journal style but may be corrected during copy editing. Likely was modified.*

Winslow, L.A., Read, J.S., Hansen, G.J., Rose, K.C. and Robertson, D.M., 2017. Seasonality of change: Summer warming rates do not fully represent effects of climate change on lake temperatures. Limnology and Oceanography, 62(5), pp.2168-2178.

---

## Author Response (AR2)

**Public justification (visible to the public if the article is accepted and published)**:
Dear authors, I have looked at the revised version, considering the earlier reviewer comments. I noted that a few reviewer comments still need to be addressed, and have some minor comments after rereading your resubmitted version. Line numbers refer to the version without track and trace.

*Thank you for your close reading of our manuscript. I have addressed the comments below and the line numbers refer to the marked up version.*

Unanswered reviewer comments:
Please still include the mathematical approach (i.e. formula?) for the Schmidt stability, meromictic stability, Brunt-Vaisala or buoyancy frequency (N) and the dimensionless lake Number in the methods section.

*There was a statement in the methods that calculations were performed with an R package, but lines 106-108 to clarify this point: "The density, Brunt-Väisälä or buoyancy frequency ($N^2$, $s^{-2}$), and the Schmidt stability were calculated using the RLakeAnalyzer package v.1.11.4.1 with equations described in that packages documentation (Winslow et al., 2019)."*

*The dimensionless lake number calculation is described in lines 208-209, but this sentence has been further clarified to make that more explicit.*

Associate editor comments:
After the discussion that describes i) mixing, ii) chemical and iii) biological characteristics, it would be good if there was a short paragraph where you can discuss how these parameters interact. I.e., how are chemical characteristics predicted by the physical mixing, and or biological characteristics predicted by the chemical characteristics? This can be done per lake, or across the lake systems. If there is no connection to be made, a few lines where you explain this can be placed in the discussion.

*To better frame these characteristics in the results & discussion, I have written a preamble for that section that describes how these interact (lines 131-138). Additionally, I have added some clarifying statements to the conclusion section to better address this integration.*

L 110, define SCML

*Thanks for catching this. The definition was originally in the introduction but that line was moved to the discussion following the recommended revisions. The definition for this acronym has been moved to line 117.*

L 119, please describe briefly what data was retrieved from the student reports.

*Where student data is used it is cited and the full citation information is in the reference list. However, this statement is now in lines 128-129: "Morphometric data, temperature and conductance profiles, observations of water column mixing, and inferences from biological experiments in these reports are referenced in the current study."*

L 238, "with the balance mostly from magnesium", the meaning of this expression is not clear to me, please rephrase.

*This is the charge balance. "Charge" has been added before balance in line 255.*

L 293, please add a few more words to explain why zooplankton grazing causes a subsurface chlorophyll maximum.

*Line 310 has been updated to: "and avoidance of zooplankton grazing in the epilimnion".*

L 295, 'that was persistently present'. Please check throughout the text that the revisions did not result in missing words.

*Present was added to line 313. The entire manuscript was carefully read and edited for clarity.*

Additional private note (visible to authors and reviewers only):
Dear author, as the requested corrections are minor, they will be reviewed by the associate editor after resubmission.

Best regards,
Cindy De Jonge